# A Common Polymorphism in *RNASE6* Impacts Its Antimicrobial Activity toward Uropathogenic *Escherichia coli*

**DOI:** 10.3390/ijms25010604

**Published:** 2024-01-03

**Authors:** Raul Anguita, Guillem Prats-Ejarque, Mohammed Moussaoui, Brian Becknell, Ester Boix

**Affiliations:** 1Department of Biochemistry and Molecular Biology, Faculty of Biosciences, Universitat Autònoma de Barcelona, 08193 Cerdanyola del Vallès, Spain; raul.anguita@uab.cat (R.A.); guillem.prats.ejarque@uab.cat (G.P.-E.); mohammed.moussaoui@uab.cat (M.M.); 2Kidney and Urinary Tract Center, The Abigail Wexner Research Institute at Nationwide Children’s Hospital, Columbus, OH 43205, USA

**Keywords:** RNase, RNase 6, RNase k6, single nucleotide polymorphisms, antimicrobial peptides, urinary tract infections, uropathogenic *Escherichia coli*

## Abstract

Human Ribonuclease (RNase) 6 is a monocyte and macrophage-derived protein with potent antimicrobial activity toward uropathogenic bacteria. The *RNASE6* gene is heterogeneous in humans due to the presence of single nucleotide polymorphisms (SNPs). *RNASE6* rs1045922 is the most common non-synonymous SNP, resulting in a G to A substitution that determines an arginine (R) to glutamine (Q) transversion at position 66 in the protein sequence. By structural analysis we observed that R66Q substitution significantly reduces the positive electrostatic charge at the protein surface. Here, we generated both recombinant RNase 6-R66 and -Q66 protein variants and determined their antimicrobial activity toward uropathogenic *Escherichia coli* (UPEC), the most common cause of UTI. We found that the R66 variant, encoded by the major SNP rs1045922 allele, exhibited superior bactericidal activity in comparison to the Q66 variant. The higher bactericidal activity of R66 variant correlated with an increase in the protein lipopolysaccharide binding and bacterial agglutination abilities, while retaining the same enzymatic efficiency. These findings encourage further work to evaluate *RNASE6* SNP distribution and its impact in UTI susceptibility.

## 1. Introduction

Bacterial urinary tract infections (UTI) afflict 150 million people annually worldwide [1], with a lifetime incidence of 50–60% in women [2], 25–30% of whom suffer recurrent UTI within six months [3]. Uropathogenic *Escherichia coli* (UPEC) is the most common cause of UTI, accounting for 80–90% of cases [1,3]. The risk of ascending UTI and pyelonephritis is elevated in young children, people with diabetes, and the geriatric population, and this can lead to acute kidney injury and chronic kidney disease due to renal scarring [4,5,6]. The choice of antibiotics to manage UTI has been limited by mounting antimicrobial resistance [7,8]. These circumstances drive the demand for novel measures to identify patients most at risk for UTI recurrence.

A greater understanding of the human immune response to UTI should yield insights into mechanisms that account for heightened susceptibility to infection along with new strategies to combat UTI. Experiments in preclinical models of UTI attest that the innate immune system is chiefly responsible for UPEC detection and clearance [9,10,11]. The innate immune system of the urinary tract relies upon a combination of pattern recognition receptors, complement activation, phagocyte recruitment, and antimicrobial peptides and proteins (AMPs) to detect and destroy invading uropathogens [10,12,13]. The antimicrobial mechanisms of AMPs include among others: membrane disruption, microbial agglutination, blockade of cell division, impaired ribosomal translation, and micronutrient sequestration [14,15,16,17]. In vitro and in vivo studies have implicated AMPs in host defense against UPEC-associated UTI. AMPs are synthesized by multiple cell types in the urinary tract, including urothelial cells, renal intercalated cells, neutrophils, monocytes, and macrophages [12,18,19].

Mounting evidence suggests that AMP levels and antimicrobial properties are reduced in patients with recurrent UTI (rUTI). Urine from girls with rUTI contains lower levels of AMPs, such as RNase 4, RNase 7, and Lipocalin-2/NGAL, compared to healthy controls [20,21,22]. Such quantitative defects may have an underlying genetic basis, as patients with fewer copies of immune related genes such as the defensin *DEFA1A3* and *DMBT1* genes experience rUTI more frequently [23,24]. Moreover, non-synonymous single nucleotide polymorphisms (SNPs) in AMP encoding genes can result in defective antimicrobial activity. Along these lines, a common SNP in *RNASE7* that encodes a Pro103 to Ala substitution (rs1263872) reduces its bactericidal activity toward UPEC and is more prevalent in children with UTI [25]. Altogether, these studies indicate promising roles for AMP levels and genotypes as new prognostic tools to identify patients at high risk for recurrence.

We previously identified RNase 6 as a monocyte and macrophage -derived AMP that is expressed in the urinary tracts of humans and mice [16]. RNase 6, also named RNase k6, was originally identified as an orthologue of bovine kidney RNase2 when tracing the evolutionary divergence within the RNaseA superfamily [26], a family that groups proteins endowed with a diversity of host defense properties [27]. By structure-functional studies we demonstrated that RNase 6 antimicrobial mechanism mostly relies on its action at the bacteria envelope and is dependent on both surface exposed hydrophobic and cationic residues [28]. The protein is more active on Gram-negative bacteria, showing a high affinity for lipopolysaccharide (LPS) [28]. Interestingly, recombinant human and mouse RNase 6 proteins exhibited bactericidal activity toward uropathogenic *Escherichia coli* (UPEC) at low-micromolar concentrations [16]. Accordingly, *RNASE 6* transgenic mice are less susceptible to UPEC induced experimental UTI than non-transgenic controls [29]. In this study, we have characterized the most common non-synonymous *RNASE6* SNP in the human population and assessed its impact on RNase 6 antimicrobial activity toward UPEC.

## 2. Results

### 2.1. RNASE6 rs1045922 Is a Common, Non-Synonymous SNP That Alters RNase 6 Antimicrobial Activity

Since previous studies have identified associations between genetic variants in antimicrobial proteins and UTI, we hypothesized that a common coding SNP in *RNASE6* may impact its antimicrobial activity toward UPEC. We performed a query at the dbSNP (https://www.ncbi.nlm.nih.gov/snp/ (accessed on 26 August 2023)) and identified rs1045922 as the most common, nonsynonymous variant in the protein coding region of *RNASE6,* with a minor allele frequency of 0.25–0.3 in the global population. An overall analysis of the G > A frequency highlights significant differences among the main geographical subareas, with a more than 75% predominance of G over A in the African population and a more balanced distribution in the East and South Asian groups (Figure 1).

The minor allele confers a G-to-A nucleotide substitution, leading to an arginine to glutamine transversion at amino acid position 66 in the mature RNase 6 protein (i.e., R66Q). To test the influence of R66Q on the antimicrobial capacity of RNase 6, we generated and purified recombinant RNase 6-R66 and RNase 6-Q66 proteins. We found that RNase 6-R66 variant exhibited increased antimicrobial activity toward laboratory and clinical cystitis and pyelonephritis strains of *E. coli* (UTI89 and CFT073) [31,32], when compared to RNase 6-Q66 (Figure 2 and Table 1).

Each assay was performed at least in triplicate. Values denote mean ± standard error of the mean (SEM).

Mechanistically, RNase 6-Q66 displayed reduced LPS binding affinity and *E. coli* agglutination (Table 2) compared to the RNase6-R66 protein. Both of these properties are essential mechanisms implicated in the antimicrobial action of RNase 6 [28].

LPS binding was assessed using the cadaverine-BODIPY TR (BC) fluorescent probe. EC_50_ indicates the protein concentration that achieves 50% effective BC displacement and “Max” refers to the maximum binding percentage (%), where 100% corresponds to total displacement and 0 corresponds to no displacement of the fluorescent dye. Three independent measurements were performed for each condition. Values denote mean ± SEM.

### 2.2. Both SNP RNase6-R66 and -Q66 Display the Same Catalytic Activity

In contrast, the Arg to Gln substitution at position 66 did not have any effect on the protein catalytic activity, as evaluated by comparison of the initial velocities toward dinucleotide substrates (Table 3). Equivalent activities were registered for both UpA and CpA, whereas no detectable activity was observed for UpG, as previously reported for RNase6-R66 [33]. Both variants retained an equivalent U/C specificity at the main substrate base site (B1) and selectivity for adenine at the secondary site (B2).

Data expressed in% activity relative to RNase 6-R66 based on mean values of triplicate assays. The substrate concentration was 100 μM and enzyme concentration was 1 μM for UpA, UpG and CpA.

### 2.3. The RNase6 R66Q Substitution Significantly Reduces the Positive Electrostatic Charge at the Protein Surface

Finally, we predicted the influence of R66Q substitution within the RNase 6 3D environment. Based on the RNase 6 crystal structure solved at atomic resolution [33], R66 is a residue located at the protein surface, where it participates along with the neighboring H67 residue in a cationic cluster that interacts with both sulphate and phosphate anions (Figure 3A) [33,34]. The 3D structure of the RNase6-Q66 variant was predicted using the *AlphaFold2* server [35]. Simulation of the impact of its substitution with a non-charged amino acid (Q66) clearly illustrated how the arginine to glutamine substitution significantly reduced the positive electrostatic charge of the surface exposed region (Figure 3B). A close inspection of the structure indicates that R66Q not only leads to the loss of net cationic charge but also prevents the residue interaction with the anionic D107 residue (Appendix A). Therefore, a significant reduction of the local surface cationic patch in the Q66 variant could diminish the protein’s affinity for the anionic components within the bacterial wall.

Thus, structural analysis supports the present experimental data showing that the SNP that encodes for R66Q substitution determines a significant reduction in its LPS binding, bacteria agglutination and overall antimicrobial activities.

## 3. Discussion

While the human genome exhibits considerable diversity particularly in its compendium of genes associated with innate immunity, the functional implications of this diversity in many cases have not been fully addressed [14,36]. Mounting evidence indicates that differences in UTI susceptibility among humans may have a genetic basis [23,24,25]. Within the RNase A superfamily, some SNPs have been associated with disease predisposition and infection susceptibility [25,37,38,39].In this study, we focused on the functional consequences of the most common, non-synonymous SNP in *RNASE6* on its antimicrobial properties toward UPEC, the most common cause of bacterial UTI.

The comparison of the antimicrobial properties of the resulting RNase 6-Q66 and RNase6-R66 variants toward UPEC strains illustrated significant differences (Figure 2 and Table 1). RNase 6 antimicrobial potency has been partly associated with its capacity to bind to LPS at the bacterial cell wall and agglutinate cells [28]. In this regard, it is noteworthy that RNase 6-Q66, the minor variant (Figure 1), was less effective in LPS binding and *E. coli* agglutination when compared to the predominant RNase 6-R66 protein (Table 2). A close structural inspection revealed that R66 contributes to a cationic region that favors anion ligand binding (Figure 3), as observed in the solved crystal structures of RNase 6 in complex with either sulphate or phosphate anions (PDB IDs: 4X09 and 5OAB) [33,34]. Indeed, the N64-R66 stretch was identified by PDBe motif as a cation region prone to bind anionic molecules, and R66 was identified as a key residue for the protein’s putative saccharide binding by molecular modelling [40]. Likewise, the cationic residues at the protein surface were identified to interact with the anionic bacterial LPS in RNase 3, another RNase A family homologue with antimicrobial properties [41]. RNase binding to LPS was correlated to the induction of bacterial agglutination by screening a battery of LPS progressively truncated *E.coli* strains [42]. Thus, we posit that decreased surface cationic charge accounts for reduced LPS binding, *E. coli* agglutination, and microbicidal activity of RNase 6-Q66.

On the other hand, R66Q substitution did not alter the enzyme catalytic activity (Table 3). Kinetic results using dinucleotides as substrates did not reveal any significant difference in the catalytic efficiency or in the enzyme base preference. Although structural data indicates that the 64–68 loop is the main anchoring region for RNase 6 binding to adenine at the B2 site, residue 66 would not interact directly with the base ring. Recent solving of RNase 6 crystal structure in complex with an adenine mononucleotide revealed direct hydrogen bonding with N64 and N68, but no direct interaction with R66 [43]. Overlapping of the predicted 3D structure of the RNase6-Q66 variant onto the RNase6-R66 in complex with AMP (PDB ID: 6MV7) [43] suggests equivalent interactions with the nucleotide, where N64 and N68 in both variants could bind to the adenine ring and the neighbor R/Q66 residues cannot make direct interactions (Appendix A). Previous structural analysis by molecular dynamics within the RNase A superfamily highlighted position 66 in RNase 6 as counterpart to Q69 in RNase A [33,44], where Q69 can complement the role of N71 (N68 in RNase 6). The previous work by molecular dynamics also highlighted the potential roles of both N64 and N68 for direct binding to adenine at B_2_ but no direct contribution to R66 [44]. Overlapping of RNase 6-AMP complex with RNase A- d(CpA) corroborates the equivalent roles of N64/N67 and N68/N71 in both RNases, but alternate orientation for R66 in RNase 6 and Q69 in RNase A (Appendix A). Besides, whereas R66 side chain in RNase 6 is determined by electrostatic interactions with D107, in the Q66, the side chain might perform equivalently to Q69 in RNase A. Therefore, further work would be needed to fully evaluate the implications of R66Q substitution on RNase 6 substrate selectivity.

Interestingly, evolutionary studies of RNase 6 lineage indicated an unusual low substitution rate in comparison to other family lineage types [45]. Among the few non-synonymous substitutions, we observe a trend for Gln to Arg substitution at position 66 from lower to higher order primates, which correlates with a slight increase in the protein predicted pI. In fact, position 66 stands out as RNase 6 lineage specific when mapping the sequence evolutionary rates among the RNase A superfamily homologues using the *Consurf* server (Appendix A) [46]. Whereas R66 is conserved in the 4 hominid species, all the old-world monkeys have a Gln at this position (Appendix A) and the new-world monkeys present significant differences at this region, with overall an average lower estimated pI [45]. Further work would be required to consider the functional significance of sequence diversity at this location.

Our query of dbSNP identified *RNASE6* rs1045922 as the most common, non-synonymous SNP in the human population. Studies are warranted to examine *RNASE6* rs1045922 genotypes in combination with other common variants in genes associated with the innate immune response in patients with UTI. Such studies will benefit from examination of additional, common SNP haplotypes within *RNASE6* and more broadly within the RNase A Superfamily, such as the *RNASE7* rs1263872 polymorphism recently associated with UTI susceptibility [25]; as it is plausible that combinations of these SNP converge to impact UTI risk.

In this study, we have identified *RNASE6* rs1045922 as a common, functionally significant SNP within the human population and implicated R66 as a key amino acid residue for the antimicrobial potency of RNase 6 toward UPEC. Further studies are required to investigate whether RNase 6-Q66 and RNase6-R66 variants exert differential antimicrobial activity toward Gram-positive bacterial uropathogens. Additional work is envisaged to consider the association of this and other *RNASE6* SNPs with UTI susceptibility.

## 4. Materials and Methods

### 4.1. Materials

Isopropyl_-D-1-thiogalactopyranoside (IPTG), was from Apollo Scientific (Bredbury, Chesire, UK). LPS from E. coli serotype 0111:B4 (type XII) were purchased from Sigma-Aldrich (St. Louis, MO, USA). BODIPY^®^ cadaverine (BC) was from Molecular Probes (Eugene, OR, USA). CpA, UpA and UpG were from IBA Life Sciences. *Escherichia coli* BL21 was purchased at *Novagen*, Madison, WI, USA). UTI89 is a clinical UPEC isolate from a patient with cystitis [31]. CFT073 is a clinical UPEC isolate from a patient with pyelonephritis and urosepsis [32]. All strains were inoculated from glycerol stocks and grown statically in LB medium for 16 h at 37 °C.

### 4.2. Expression of RNase6-R66 and -Q66 Variants

The RNase 6-Q66 variant was generated by site-directed mutagenesis [28]. Recombinant proteins were expressed and purified from inclusion bodies as previously described [28]. Briefly, the genes were subcloned into the plasmid pET11c for prokaryote high yield expression in an *E. coli* BL21(DE3) strain. Bacteria were grown in Terrific broth (TB), containing 400 μg/mL ampicillin. Recombinant protein was expressed after cell induction with 1 mM IPTG added when the culture showed an OD600 of 0.6. The cell pellet was collected after 4 h of culture at 37 °C. Following bacteria cell lysis and solubilization of inclusion bodies, the protein was then refolded for 72 h at 4 °C by a rapid 100-fold dilution into 100 mM Tris/HCl, pH 8.5, 0.5 M of guanidinium chloride, and 0.5 M L-arginine, and oxidized glutathione (GSSG) was added to obtain a DTT/GSSG ratio of 4. The folded protein was then concentrated, buffer-exchanged against 150 mM sodium acetate, pH 5, and purified by cation-exchange chromatography on a Resource S (GE Healthcare) column equilibrated with the same buffer. The protein was eluted with a linear NaCl gradient from 0 to 2 M in 150 mM sodium acetate, pH 5.

### 4.3. Minimum Bactericidal Concentration (MBC) Determination

Minimal Bactericidal Concentration (MBC) was determined as previously [47]. Briefly, an exponential phase bacterial subculture adjusted to 5 × 10^5^ CFU/mL in Hepes Buffer Saline (HBS) was incubated with recombinant RNase serially diluted from 20 to 0.31 μM during 4 h at 37 °C in mild agitation (100 rpm). Treatments were performed in a 96 well plate in 100 μL of volume. After incubation, 30 μL of the wells content is seeded into LB Petri dishes and incubated overnight at 37 °C for colony counting.

### 4.4. Lipopolysaccharide Binding Assay

Lipopolysaccharide (LPS) binding was determined using the cadaverine-BODIPY TR (BC) fluorescence assay, where BC displacement was monitored. Serial protein dilutions were prepared in a 96-wells fluorescence plate from 20 µM, in HEPES 10 mM pH 7.4. Then, LPS (10 µg/mL) and BC (10 µM) were added in each well and the fluorescence was read on a Victor3 plate reader (PerkinElmer, Waltham, MA, USA) with an excitation wavelength of 580 nm and 620 nm of emission. Fluorescence of free BC was registered and the binding to LPS was calculated as previously [48].

### 4.5. Bacterial Agglutination Assay

Bacterial agglutination was determined by calculation of the Minimal Agglutination Concentration (MAC) as previously [28]. Briefly, serial dilutions of the proteins were prepared on a 96-well ELISA plate in PBS, starting at 10 μM. Negative controls containing only buffer instead of protein, and bacteria were added to all wells with a final OD600 of 0.2. Then, the plates were incubated for 4 h at 37 °C and the bacterial aggregates were observed using a binocular stereo microscope at 50×. The MAC was defined as the lowest concentration where agglutinates could be seen. Three independent repeats of each assay were performed.

### 4.6. Spectrophotometric Kinetic Assay

Enzymatic activity was assayed by spectrophotometry as previously described [33]. Briefly, dinucleotides (IBA Life Sciences, Göttingen, Germany) were used as substrates and assays were carried out in 50 mM sodium acetate and 1 mM EDTA, pH 5.5, at 25 °C, using 1-cm path length cells. The activity was measured by following the initial reaction velocities using the difference molar absorbance coefficients, in relation to the cleaved phosphodiester bonds. The relative activity was calculated by comparison of initial velocities (V_0_).

### 4.7. 3D Structure Modelling

The R66Q model was predicted using the *AlphaFold2* server [49] based on the solved 3D structure of RNase 6-R66 [33]. Five equivalent output models were generated with more than 90% liability over the whole protein sequence.

## Figures and Tables

**Figure 1 ijms-25-00604-f001:**
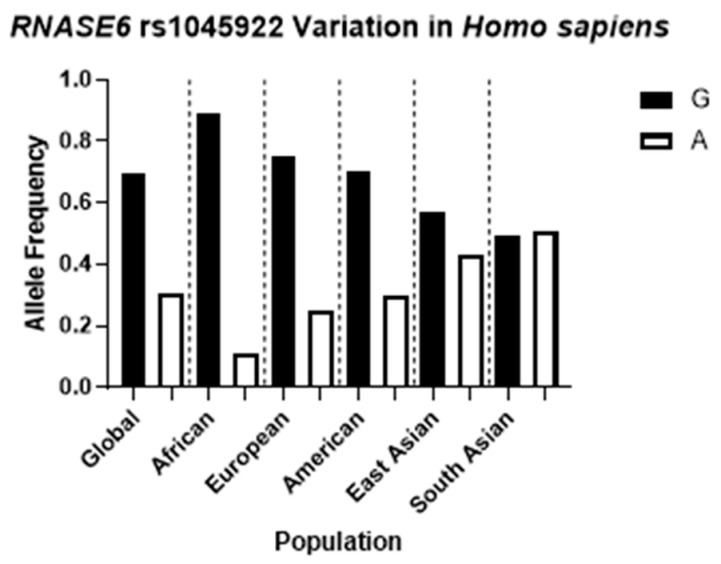
*RNASE6* rs1045922 SNP distribution among populations according to the 1000 Genomes Study [30].

**Figure 2 ijms-25-00604-f002:**
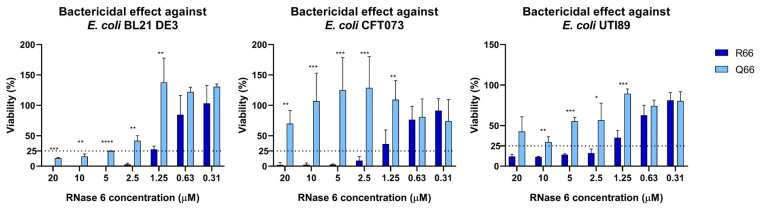
RNase 6-Q66 exhibits reduced antimicrobial activity toward laboratory and uropathogenic strains of *E. coli*, when compared with RNase 6-R66. Bacterial viability was performed by CFU counting taking the non-treated control as a 100% reference. Significant difference between the variants at each concentration is indicated (**** *p* < 0.0001; *** *p* < 0.0002; ** *p* < 0.002; * *p* < 0.03). The table below indicates the calculated Minimum Bactericidal Concentration (MBC) at which 100 or 75% of bacteria was eradicated. Horizontal dotted lines indicate the bacterial viability level taken to calculate MBC_75_ values.

**Figure 3 ijms-25-00604-f003:**
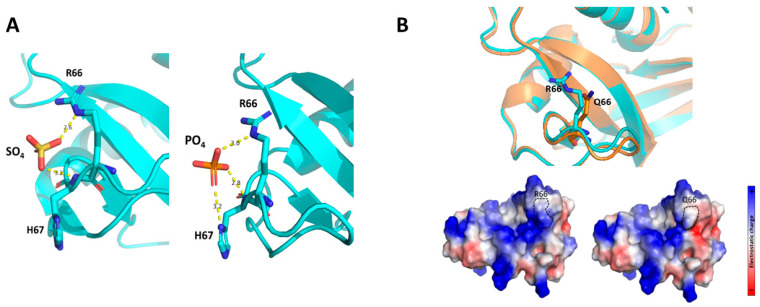
Analysis on the impact of the R66Q substitution on the protein structure. (**A**) Details of sulphate and phosphate binding interactions of RNase 6-R66 in solved crystal structures (PDB ID: 4X09 and 5OAB respectively). RNase 6 main chain is drawn in cyan ribbon. Ligands and protein interacting side chains are drawn in ball-and-sticks and colored according to atom elements (**B**) The substitution of an arginine by a non-charged amino acid at the protein surface alters the cationic charge of this region, which in turn may disturb the ability of RNase 6 to bind to anionic bacterial wall components. The mutant structural prediction was obtained by *AlphaFold2*. Top: RNase 6-R66 is shown in cyan while the -Q66 variant is shown in orange. Bottom: Surface electrostatic charge prediction. The position corresponding to the amino acid change is indicated by the dotted black line. Pictures were drawn with *PyMol* 2.3.4.

**Table 1 ijms-25-00604-t001:** Minimum Bactericidal Concentration (MBC) values.

	*E. coli* BL21		*E. coli* CF073		*E. coli* UTI89	
	MBC_100_	MBC_75_	MBC_100_	MBC_75_	MBC_100_	MBC_75_
**RNase 6-R66**	8.33 (±2.9)	2.08 (±0.72)	>20	1.87 (±0.68)	>20	3.33 (±1.44)
**RNase 6-Q66**	>20	8.22 (±2.9)	>20	>20	>20	>20

**Table 2 ijms-25-00604-t002:** LPS binding and Minimal Agglutination Concentration (MAC) for RNase 6 variants toward *E. coli*.

	LPS Binding		MAC
	EC_50_ (μM)	Max (%)	(µM)
**RNase 6-R66**	2.64 ± 0.23	75.90 ± 4.41	0.22 ± 0.05
**RNase 6-Q66**	5.15 ± 2.8	29.71 ± 10.5	1.38 ± 0.24

**Table 3 ijms-25-00604-t003:** Comparison of the relative catalytic activity of the two RNase 6 variants.

	UpA	UpG	CpA
**RNase 6-R66**	100	^ND^	100
**RNase 6-Q66**	87.5	^ND^	92

^ND^ Not Detected at the assayed conditions.

## Data Availability

Any data supporting the reported results can be provided upon request.

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
