# Peer review of "A Common Polymorphism in RNASE6 Impacts Its Antimicrobial Activity toward Uropathogenic Escherichia coli"

_ijms, 2024, doi:10.3390/ijms25010604_

Round 1
Reviewer 1 Report
Comments and Suggestions for Authors
The present manuscript "A Common Polymorphism in RNASE6 Impacts its Antimicrobial Activity Toward Uropathogenic Escherichia coli" is focused on the evaluation of Human Ribonuclease (RNase) 6 against uropathogenic bacteria. The study is well-conducted and clearly exposed. In the introduction, the authors shoud describe why they choice E.coli as model, and if they evaluated also other bacterial strains responsible of urinary infections.
Author Response
We thank the reviewer for our manuscript revision. Regarding the reviewer remark, we state in the introduction (lines 35-36) that “Uropathogenic Escherichia coli (UPEC) is the most common cause of UTI, accounting for 80-90% of cases”, and the ensuing paragraphs present introductory material about the host innate immune response during UPEC associated UTI. We have modified the revision to highlight that AMPs have been implicated in host defense in preclinical models of UPEC-associated UTI (line 52). In the conclusion, we have added the statement that “Further studies are required to investigate whether RNase 6-Q66 and RNase6-R66 variants exert differential antimicrobial activity toward Gram-positive bacterial uropathogens” (lines 235-237).
Reviewer 2 Report
Comments and Suggestions for Authors
Dear authors,
UTI are important community-aquired and hospital-aquired infection. Increased antimicrobial resistence to common UTI pathogens opens a lot of issues about proper treatment of UTI as well as about mechanisms which are inherited and thus patients are prone to this specific disease. I have read your results with a big interest and they seems to be a good beginning in a long way from laboratory findings to appropriate clinical diagnostic procedures and early treatment. I recommend your work to be published.
Author Response
We heartly thank the reviewer for the positive evaluation of our work.
Reviewer 3 Report
Comments and Suggestions for Authors
The authors presented a very interesting document entitled "A common Polymorphism in RNASE6 Impacts its Antimicrobial Activity Toward Uropathogenic Escherichia coli". The paper is well-written with an adequate scientific sound, additionally propose an alternative treatment in antimicrobial resistance age.
The methodology used are the necessary for this kind of studies and the conclusions are supported by their findings.
I only have minor comments listed below:
1. There are some correction marks across the document (e.g. lines 39, 52, 54). I suppose it was for the change tracking of the text editor. Please verify.
2. Line 52. Why is highlighted "UPEC-associated" text?
3. Figure 2 Involve a table, that could be included as table instead. Please consider changing and use the text font type for the table.
4. Table 1. The table title must be at the top (e.g. LPS binding... E. coli). The remaining text must be as foot table. Please format the table using the text font type or the template of the journal.
5. Table 2. Please format it with the text font type.
6. Line 168. Why is highlighted the word "fully"?
Author Response
We appreciate the positive revision of our work. We indicate below the specific changes applied to our manuscript according to reviewer comments: 1-Marks related to edition tracked changes have been deleted.2-Highlighted text is related to tracked changes in response to reviewer 1.3-We have edited figure 2 table as a separate table (now Table 1). Table font style now follows the same font type as main text.4-We have edited the table legend to separate accordingly its title and footnote. Table font style has also been revised.5-We also formatted Table 2 (now Table 3) with text font style.6-Highlighted word is related to tracked changes in response to reviewer 1.